# Fathom-DeepResearch: Unlocking Long Horizon Information Retrieval and Synthesis for SLMs

**Shreyas Singh\***
Fractal AI Research
shreyas.singh@fractal.ai

**Kunal Singh\*†**
Fractal AI Research
kunal.singh@fractal.ai

**Pradeep Moturi\***
Fractal AI Research
pradeep.moturi@fractal.ai

## ABSTRACT

Tool-integrated reasoning has emerged as a key focus for enabling agentic applications. Among these, DeepResearch Agents have gained significant attention for their strong performance on complex, open-ended information-seeking tasks. We introduce Fathom-DeepResearch, an agentic system composed of two specialized models. The first is Fathom-Search-4B, a DeepSearch model trained from Qwen3-4B and optimized for evidence-based investigation through live web search and targeted webpage querying. Its training combines three advances: (i) DUETQA, a 5K-sample dataset generated via multi-agent self-play that enforces strict web-search dependence and heterogeneous source grounding; (ii) RAPO, a zero-overhead extension of GRPO that stabilizes multi-turn Reinforcement Learning with Verifiable Rewards through curriculum pruning, reward-aware advantage scaling, and per-prompt replay buffers; and (iii) a steerable step-level reward that classifies each tool call by cognitive behavior and marginal utility, enabling explicit control over search trajectory breadth, depth, and horizon. These improvements enable reliable extension of tool-calling beyond 20 calls when warranted. The second is Fathom-Synthesizer-4B, trained from Qwen3-4B, which converts multi-turn DeepSearch traces into structured, citation-dense DeepResearch Reports for comprehensive synthesis. Evaluated on DeepSearch benchmarks (SimpleQA, FRAMES, WebWalker, Seal0, MuSiQue) and DeepResearch-Bench, the system achieves state-of-the-art performance in the open-weights category while closely rivaling proprietary closed systems, while also demonstrating strong performance in general reasoning benchmarks: HLE, AIME-25, GPQA-Diamond, and MedQA.

⟩https://github.com/FractalAIResearchLabs/Fathom-DeepResearch

## 1 INTRODUCTION

Recent advancements in reasoning capabilities of Large Language Models (LLMs) have enabled a significant performance advancement across a diverse set of tasks, such as mathematical reasoning, code generation (Jain et al., 2024; DeepSeek-AI et al., 2025; Singh et al., 2025b;a). We are not only witnessing expert level performance on academic benchmarks, but are perceiving a paradigm shift towards agentic intelligence. Owing to tool-integrated reasoning, these models can now autonomously observe, reason and interact with complex and dynamic environments. Contemporary state-of-the-art tool-augmented AI systems like DeepResearch (OpenAI, 2025b; Team, 2025a;c), have exhibited super-human performance in highly sophisticated long-horizon, deep-information retrieval and synthesis tasks. These agents transcend the limitations of static parametric knowledge by

---

*Equal contribution.
†Project lead.

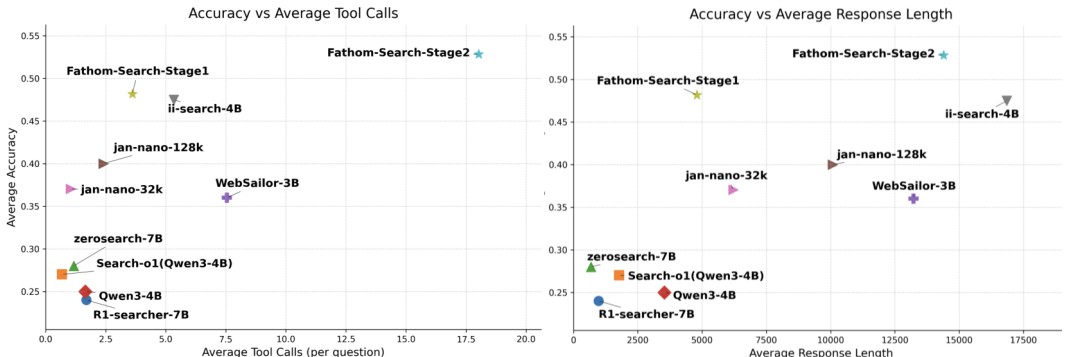

Figure 1: **Accuracy vs Response length** & **Accuracy vs Avg.Tool calls** plot comparing open-source DeepSearch models, clearly demonstrates the higher accuracy and efficient long horizon tool interaction ability of Fathom-Search models compared to its contemporaries.

implementing dynamic reasoning frameworks that autonomously partition multifaceted queries, coordinate multiple tool interactions, and integrate heterogeneous information sources into unified, evidence-supported conclusions.

However, a substantial performance gap remains between (OpenAI, 2025b; Team, 2025a;c) and open-source (Team, 2025b; AI, 2025), making the development of robust DeepResearch architectures a critical challenge. Current open-source frameworks suffer from two fundamental limitations. First, they lack proficiency in sustained tool usage required for high-uncertainty reasoning and synthesis tasks (Wu et al., 2025; Pham et al., 2025; Trivedi et al., 2022; Krishna et al., 2024; Wei et al., 2024). Efforts to scale DeepResearch capabilities are constrained by (i) the absence of a high-quality, verifiable, and scalable dataset creation pipeline, (ii) algorithmic instability in multi-turn reinforcement learning (RL) with tools, and (iii) inefficient tool-calling behavior that undermines deep information exploration and retrieval. Second, there is an overemphasis on closed-form problem solving, which comes at the expense of the information synthesis capabilities essential for tackling open-ended investigative queries. In the next section we discuss the aforementioned issues further.

## 1.1 MOTIVATION

**(1) Training instability of GRPO in multi-turn tool interaction**: RLVR (Reinforcement Learning with verifiable rewards) with GRPO (Shao et al., 2024) has demonstrated early promise in aligning LLMs with sparse reward signals for single-turn reasoning tasks, particularly in structured domains like Math/STEM Shao et al. (2024); Yang et al. (2024). However, GRPO struggles to scale to multi-turn tool-augmented environments, because external tool interaction responses induce distribution shift in the policy model from its set token generation patterns, this leads to decoding instability and malformed generations. This cascading of errors causes group-relative advantages to saturate, leading to extremely unstable gradient updates that breaks the entire training process. (Xue et al., 2025).

**(2) Reward hacking and inefficient tool calling** *(a) Correctness-only sparse rewards do not scale to long-horizon tool calling.* When training with only a single end of episode correctness signal, the agent shows early improvements achieving format adherence and basic tool-calling competence in the beginning, however, as training progresses, tool usage increases sharply while both training reward and validation performance deteriorate (Nguyen et al., 2025). This degradation stems from reward hacking: the agent collapses into repetitive, identical tool calls because the vanilla RLVR objective provides no incentive for efficiency or diversity in tool use. *(b) RL amplifies SFT priors, limiting control over the cognitive behaviors developed by the policy* (Gandhi et al., 2025): Tool-use RL typically relies on an SFT cold start to elicit basic tool competence (Li et al., 2025a); (Dong et al., 2025) RL then amplifies pre-existing cognitive behaviors seeded by SFT. Standard RLVR affords limited control over the exploration and verification strategies developed by the policy model, consequently the quality of cold-start trajectories disproportionately shape the policy model's tool-use behavior and provides no steerability.

**(3) Limited training data characterized by high and hard-to-reduce intrinsic information uncertainty**: Training datasets such as TriviaQA (Joshi et al., 2017), and multi-hop variants like

2WIKI(Ho et al., 2020), and HotpotQA (Yang et al., 2018) represent problems where solutions can often be found through minimal set queries or even from a model's parametric knowledge alone. These datasets do not expose models to the real-world retrieval challenges posed by noisy, heterogeneous data sources on the internet. Recent synthetic efforts (Sun et al., 2025a; Li et al., 2025a; Sun et al., 2025b) attempt to bridge this gap by simulating realistic search behavior. For instance, WebSailor's(Li et al., 2025a) SailorFog-QA constructs ambiguous queries using obfuscated subgraphs of entity graphs, while SimpleDeepResearcher (Sun et al., 2025b) issues multi-stage search-summarize-generate tool calls over raw HTML. Despite their innovation, these pipelines remain expensive, brittle, and time-consuming. They rely on handcrafted heuristics, graph expansion, or multi-stage LLM orchestration, limiting scalability, topical diversity, and adaptability to new domains.

**(4) Challenges in handling open-ended queries** Recent work largely targets closed-ended queries with well-defined objectives (Dao & Vu, 2025; Internet, 2025; Li et al., 2025a). In contrast, many real-world tasks are open-ended: they lack a single definitive answer and require multi-turn exploration, retrieval of diverse perspectives, and synthesis of evidence-grounded conclusions. Addressing these challenges, demands strong information-synthesis and verification capabilities, gaps that current open-source approaches largely leave unfilled.

## 1.2 OUR CONTRIBUTIONS

To this end, we present an end-to-end DeepSearch system centered on *Fathom-Search-4B* (search enabled reasoning) and *Fathom-Synthesizer-4B* (synthesis & report-generation). Our key contributions:

- **RL Zero framework for DeepSearch training.** We present a novel two-stage RL-Zero framework that helps to *steer cognitive behaviors* developed by the policy model like exploration & verification during the training.

- **RAPO: Reward Aware Policy Optimization.** We introduce a zero-overhead modification of GRPO with *dataset pruning, advantage scaling, and replay buffers, and a steerable step-level reward* that stabilizes multi-turn RL and enables long-horizon tool use.

- **DUETQA.** We make open-source release of 5K sample dataset created through our novel *multi-agent self-play pipeline*, which has verifiable question-answer pairs, impossible to answer without *live web search* from heterogeneous sources, unconstrained on any specific domains (e.g. wikipedia) for DeepSearch model training.

- **DEEPRESEARCH-SFT.** A synthetic SFT corpus for converting downstream search/investigation traces of DeepSearch enabled models into comprehensive citation-backed DeepResearch reports via an explicit *plan then write* protocol.

- **FATHOM-4B.** We opensource **Fathom-Search-4B**, trained using RAPO policy on RL-Zero framework with DuetQA dataset, optimized for long-horizon, evidence-seeking through live web search. **Fathom-Synthesizer-4B**, finetuned with DeepResearch-SFT dataset designed for open-ended synthesis and report generation.

## 2 FATHOM-SEARCH-4B

We describe the methodology underlying *Fathom-Search-4B*, a tool-using LLM that leverages live web-search capabilities to do evidence based reasoning in a multi-turn tool interaction setting, unlocking long-horizon tool use ($> 20$ calls) ability. These capabilities arise from a combined approach of: (i) a curated synthetic data pipeline tailored to search-tool augmented reasoning, (ii) targeted upgrades to GRPO to effectively adapt it to multi-turn tool interaction, and (iii) a two-stage training regimen with reward shaping to expand the tool-use horizon in a steerable manner.

## 2.1 DUETQA: A DEEPSEARCH DATASET, GENERATED VIA MULTI-AGENT SELF PLAY

To address the aforementioned dataset challenges in (Sec. 1.1), we develop a self-supervised dataset construction framework designed to yield verifiable, search-dependent, multi-hop QA pairs. This pipeline serves as the basis for generating DUETQA, a dataset tailored for training agentic deepsearch models. The design goals are: **Live web-search dependency**: for each QA pair $(q, a)$, the question

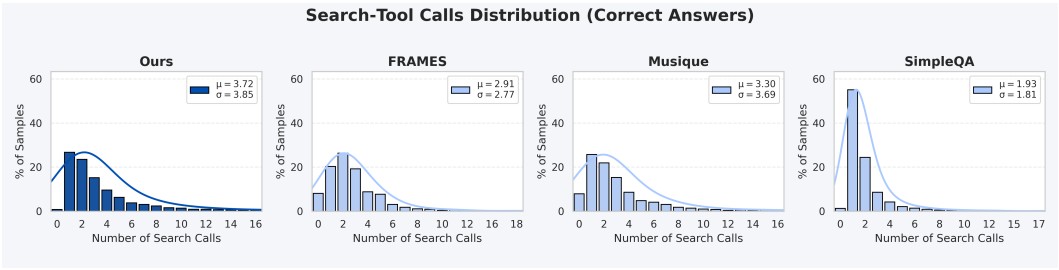

Figure 2: Distribution of number of search-calls issued by o3(OpenAI, 2025a) over correctly answered questions comparing DuetQA to other prominent benchmarks. **DuetQA shows strict live-web-search dependence and multi-hop reasoning** as evident from the long-tailed distribution (unlike simpleQA (Wei et al., 2024)) and $\geq 1$ search call(s) required to answer all DuetQA questions correctly (unlike FRAMES (Krishna et al., 2024), Musique(Trivedi et al., 2022)).

is unanswerable without search by enforcing that at least one hop contains information post–2024 (i.e., for a model $\mathcal{M}$, $P(a \mid q, \mathcal{M}_{\text{no-search}}) \ll P(a \mid q, \mathcal{M}_{\text{search}})$); **Diverse source domains**: questions require querying hetrogeneous web-sources beyond Wikipedia ; and **Steerable theme control**: each example is grounded in $k \in [5, 7]$ sampled themes from $\mathcal{T}$, a manually curated taxonomy of $200+$ themes covering a broad range of topics. We generate questions using two frontier web search enabled LRMs, $\mathcal{M}_1$ (O3) and $\mathcal{M}_2$ (O4-mini) (OpenAI, 2025a), acting as *proxy web-crawling agents* that produce QA pairs and as *independent verifiers* to that ensure question solvability; a third model, $\mathcal{M}_3$ (GPT-4o), is a *non-search* model used for controlled paraphrasing/obfuscation of questions and as a baseline verifier without search. Refer to (Appendix A.2 for more details on the dataset)

**Data Generation.** We adopt two strategies to synthesize multi-hop, search-dependent question-answer pairs. In both, we sample a set of themes $\mathcal{T}_{\text{sample}} \sim \text{Uniform}(\mathcal{T})$ with $|\mathcal{T}_{\text{sample}}| = k$, $k \in \{5, 6, 7\}$. In the *Mixture of Themes* setting, for each $t \in \mathcal{T}_{\text{sample}}$, the generator ($\mathcal{M}_1$ or $\mathcal{M}_2$) issues live queries to retrieve recent and/or obscure facts, and composes a multi-hop pair $(q, a)$ by chaining a subset of them into a coherent reasoning path. In the *Seeded Question* setting, we maintain a seed bank of 100 questions. *(50 % manually curated and 50% sampled from BrowseComp (Wei et al., 2025)*; given a seed $q_0$, the generator rewrites it into a new question $q$ by integrating one or more sampled facts while preserving the multi-hop scaffold of $q_0$. In both settings, we enforce that at least one incorporated fact references information after 2024. Refer to (Fig. 8 & Fig. 9) in Appendix for examples.

**Data obfuscation.** To remove surface cues that enable *short-circuiting* the multi-hop reasoning, we run an obfuscation pass, using $\mathcal{M}_3$ (GPT-4o) with in-context examples, we paraphrase questions, masking intermediate hops. Specifically, $\mathcal{M}_3$ attenuates exact anchors by (i) coarsening dates (e.g., "March 2025" $\rightarrow$ "early 2025"), (ii) mapping precise numerics to qualitative magnitudes (e.g., "1%" $\rightarrow$ "negligible"), (iii) replacing named entities with indirect descriptors (e.g., "University of Florida" $\rightarrow$ "a major southeastern university").

**Multi-agent Verification.** We retain a candidate pair $(q, a)$ only if two independent search-enabled LRMs $\mathcal{M}_1$ and $\mathcal{M}_2$ produce the same correct answer while a strong non-search baseline $\mathcal{M}_3$ fails. This filter enforces correctness through cross-model agreement and certifies that web retrieval is indispensable, ensuring the non-triviality of the question while guarding against overlap of information with the model's parametric knowledge.

## 2.2 AGENTIC REINFORCEMENT LEARNING

In this section, we formulate multi-turn, tool-augmented RL with LLM policies. Let $x \in \mathcal{X}$ be an input from distribution $\mathcal{D}$ and $\mathcal{T}$ the set of available tools. The policy $\pi_\theta$ generates a reasoning trajectory $\mathcal{R}$ interleaved with tool feedback, followed by a final textual answer $y$. A reference policy $\pi_{\text{ref}}$ is used for KL regularization, and a verifiable reward function $r_\phi$ (LLM-as-judge) provides supervision. The joint rollout can be written as:

$$P_\theta(\mathcal{R}, y \mid x; \mathcal{T}) = \Big[ \prod_{t=1}^{t_\mathcal{R}} P_\theta(\mathcal{R}_t \mid \mathcal{R}_{<t}, x; \mathcal{T}) \Big] \cdot \Big[ \prod_{t=1}^{t_y} P_\theta(y_t \mid y_{<t}, \mathcal{R}, x; \mathcal{T}) \Big], \quad \mathcal{R}_t = (\varphi_t, c_t, o_t), \tag{1}$$

where $\varphi_t$ is a latent "think" segment, $c_t \in \mathcal{T}$ a tool call (with arguments), and $o_t$ the tool response, all expressed in a ReAct-style template. We optimize the policy model with a token-level clipped loss defined as follows:

$$\mathcal{L}_{\text{GRPO}} = \frac{1}{G} \sum_{i=1}^{G} \frac{1}{T_i} \sum_{t=1}^{T_i} \min\Big[ r_{i,t} \hat{A}_{i,t}, \ \text{clip}(r_{i,t}, 1-\epsilon, 1+\epsilon) \hat{A}_{i,t} \Big], r_{i,t} = \frac{\pi_\theta(o_{i,t} \mid x, \mathcal{H}_{t-1})}{\pi_{\theta_{\text{old}}}(o_{i,t} \mid x, \mathcal{H}_{t-1})} \tag{2}$$

The trajectory-level scalar reward combines a format score and an answer score (Li et al., 2025a):

$$r_i = 0.1 * R_i^{\text{format}} + 0.9 * R_i^{\text{answer}} \tag{3}$$

Here, $R_i^{\text{format}}$ verifies that rollout follows the ReAct template (i.e., all steps are correctly wrapped in `<think>`, `<tool_call>`, `<tool_response>` tags). Meanwhile, $R_i^{\text{answer}} = \mathbf{1}[a_{\text{pred}}^{(i)} = a_{\text{gt}}]$, where correctness of the final answer is judged by an LLM-as-judge against the ground truth.

For a group of $G$ sampled rollouts with scalar rewards $\{r_i\}$, group-relative advantages defined as:

$$\hat{A}_{i,t} = \frac{r_i - \mu_R}{\sigma_R}, \quad \mu_R = \frac{1}{G} \sum_{j=1}^{G} r_j, \quad \sigma_R = \sqrt{\frac{1}{G} \sum_{j=1}^{G} (r_j - \mu_R)^2}. \tag{4}$$

## 2.3 RAPO: REWARD-AWARE POLICY OPTIMIZATION

RAPO is a lightweight extension of GRPO designed to stabilize multi-turn, tool-augmented training by addressing the issues outlined in Sec. 1.1. In GRPO, the per-prompt (group) reward variance $\sigma_R$ (Eq. 4) determines the strength of the advantage signal. Let $\sigma_R$ denote the within-group reward standard deviation; a group is **Good** if $\sigma_R > 0$ and **Bad** if $\sigma_R = 0$. *Bad* groups arise under both prompt saturation (all rollouts succeed) and cascading errors (all fail). In both cases, no advantage signal is produced, shrinking gradient norms and destabilizing updates. (Fig. 10 in Appendix). RAPO counters these effects through three modifications applied on top of GRPO, all at zero additional rollout cost:

**Dataset pruning.** We prune prompts solved at epoch end using $\text{SolveRate}(q) = \frac{1}{G} \sum_{i=1}^{G} \mathbf{1}[R_i > 0]$ and drop $q$ when $\text{SolveRate}(q) \geq 0.9$. This prevents training batches from being dominated by saturated groups that provide negligible variance, while implicitly yielding a curriculum in which the active set concentrates on harder prompts.

**Advantage scaling.** To counter gradient dilution when only a few groups in a batch are informative, we rescale token-level advantages for Good groups inversely with their batch frequency : $\tilde{A}_{i,t} = \frac{G}{G_{\text{good}}} \hat{A}_{i,t}$ with $G_{\text{good}} = \#\{\text{groups with } \sigma_R > 0\}$. This adjustment preserves effective gradient magnitude without requiring costly re-sampling as in DAPO (Yu et al., 2025), ensuring that updates remain stable even when informative groups are sparse.

**Replay buffer.** We maintain a per-prompt buffer $\mathcal{B}$ containing the most recent successful trajectory $\mathbf{o}^\star$ with $R(q, \mathbf{o}^\star) > 0.5$. If all rollouts for a prompt fail in the current epoch, one trajectory is randomly replaced with $\mathbf{o}^\star$ from $\mathcal{B}$. This reintroduces variance ($\sigma_R > 0$) into otherwise collapsed groups, restores group-relative advantages, and anchors updates to a high-quality, low-entropy reference that curbs uncontrolled trajectory growth.

## 2.4 STEERABLE STEP-LEVEL REWARD DESIGN FOR SEARCH TOOLS

We design our novel *Steerable Step-Level Reward* that alleviates the reward-hacking challenge faced by RLVR training in the multi-turn, tool-interaction setting using vanilla reward (Eq. 3) as described in (Sec. 1.1). Our reward function enables us to steer (i) *how much* the agent uses tools and (ii) *how* it allocates cognition to exploration and verification. Starting from the vanilla RLVR objective in (Eq. 3), we make the correctness branch $R_i^{\text{answer}}$ depend on cognitive behaviors and marginal utility aware

labels assigned to each call $c_t$ in the rollout $\mathcal{R} = \{(\varphi_t, c_t, o_t)\}_{t=1}^T$ by a GPT-4.1 LLM-as-judge as follows. Refer to (Appendix. A.1) for exact details on the search & browsing tool implementation.

```
search_urls ∈ {
    UNIQUESEARCH: (semantically new query about previously unseen entities/facts),
    REDUNDANTSEARCH: (Highly similar to a prior query; overlapping results)}
query_url ∈ {
    EXPLORATION: (first query of a new URL),
    VERIFICATION: (cross-source check on a new URL for an existing query; allowed Bᵥ times),
    REDUNDANTQUERY: (further checks for a query/fact on new URLs beyond Bᵥ)}
```

From the LLM-as-Judge tool call classification we form tallies [3] and define the following aggregates:

$$\rho = \frac{n_{\text{redS}} + n_{\text{redQ}}}{T}, \qquad \Delta_S = n_{\text{uniqS}} - n_{\text{redS}}, \qquad \Delta_Q = n_{\text{uniqQ}} - n_{\text{redQ}}. \tag{5}$$

Using these aggregates we define our *Steerable Step-Level Reward* as:

$$r_i = \begin{cases} 0.1 * R_i^{\text{format}} + \max\big((1-\rho), 0.5\big), & \text{if } a_{\text{pred}}^{(i)} = a_{\text{gt}}, \\ 0.1 * R_i^{\text{format}} + c_1 * \min\big(1, \frac{\Delta_S}{C_S}\big) + c_2 * \min\big(1, \frac{\Delta_Q}{C_Q}\big), & \text{if } a_{\text{pred}}^{(i)} \neq a_{\text{gt}}. \end{cases} \tag{6}$$

Here, $\rho$ penalizes redundant tool calls *even when the rollout is correct*, pushing for efficiency; whereas $\Delta_S$ and $\Delta_Q$ provide credit to *incorrect* rollouts that exhibit genuine, non-redundant exploration & information seeking behavior.

**Monotonocity.** We set $c_1 = c_2 = 0.2$, to ensure any incorrect rollout has $r_i \leq 0.5$, while any correct rollout has $r_i \geq 0.5$, which ensures incorrect trajectories never get rewarded more than the correct ones. $c_1 = c_2$ also ensures equal weight to `search_urls` ($\Delta_S$) and `query_url` ($\Delta_Q$).

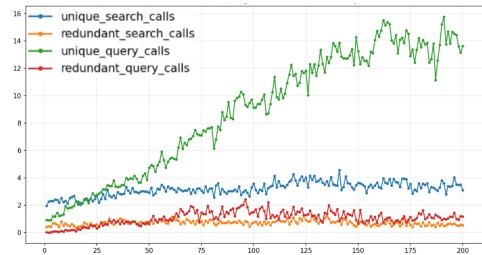

Figure 3: Evolution of unique/redundant tool-calls during Stage-2 training using our Steerable Step-level Reward (Eq.6)

**Steerability.** We expose three primary knobs: (i) $C_S$ and (ii) $C_Q$ set the saturation thresholds for creditable novelty in `search_urls` and `query_url`, respectively. Increasing $C_S$ and/or $C_Q$ raises the novelty caps, enabling more steps to earn credit when they introduce genuinely new evidence; decreasing them compresses trajectories. (iii) The per-claim verification budget $B_v$ controls verification depth: higher $B_v$ permits multiple creditable cross-checks per claim, promoting verification. For our experiments we set $B_v = 1$ allowing 1 cross-check per claim, additionally we set $C_S = 8$ and $C_Q = 16$.

## 2.5 TRAINING RECIPE

We build our reinforcement learning with verifiable rewards (RLVR) framework on top of RE-CALL (Chen et al., 2025). For web search, we use the Serper API (Serper.dev), and implement a retrieval toolchain leveraging Jina-AI together with open-source components such as TRAFILTURA and CRAWL4AI. Training is carried out in two stages. **Stage 1.** We train with RAPO for 10 epochs on our curated DUETQA dataset, comprising **4,889** high-quality QA instances. The setup uses a constant learning rate of $1 \times 10^{-6}$ with the Adam optimizer ($\beta_1 = 0.9$, $\beta_2 = 0.95$), batch size 32, mini-batch size 16, 5 rollouts per group, and top-$p = 1.0$ sampling. Each rollout is capped at 32 tool-interaction steps, with each step limited to 8,192 output tokens. The vanilla reward (Eq. 3) with $\alpha = 0.1$ is used to instill correct tool-calling behavior and strict format adherence. **Stage 2.** We continue RLVR training for an additional 2 epochs under the same hyperparameter settings. For Stage 2, we construct a mixed dataset by combining DUETQA, with math data from S1 dataset (Muennighoff et al., 2025), and the training split of MUSIQUE (Trivedi et al., 2022) and medical reasoning data from

---

[3] $n_{\text{uniqS}} \leftrightarrow \#$ UNIQUESEARCH calls; $n_{\text{redS}} \leftrightarrow \#$ REDUNDANTSEARCH calls;
$n_{\text{explore}} \leftrightarrow \#$ EXPLORATION calls; $n_{\text{verify}} \leftrightarrow \#$ VERIFICATION calls; $n_{\text{uniqQ}} = n_{\text{explore}} + n_{\text{verify}}$;
$n_{\text{redQ}} \leftrightarrow \#$ REDUNDANTQUERY calls;

Table 1: Accuracy(%) of *Fathom-Search-4B* on DeepSearch benchmarks SimpleQA, FRAMES, WebWalker, Seal0, Musique and general reasoning benchmarks HLE, AIME-25, GPQA-D, MedQA. 'Avg' is the unweighted mean within each block. Bold/italics denote best/second-best per benchmark.

| Model | DeepSearch Benchmarks | | | | | | General Reasoning Benchmarks | | | | |
| --- | --- | --- | --- | --- | --- | --- | --- | --- | --- | --- | --- |
| | SimpleQA | FRAMES | WebWalker | Seal0 | Musique | Avg | HLE | AIME-25 | GPQA-D | MedQA | Avg |
| *Closed-Source Models* | | | | | | | | | | | |
| GPT-4o (without search) | 34.7 | 52.4 | 3.2 | 7.2 | 34.0 | 26.3 | 2.3 | *71.0* | *53.0* | *88.2* | 53.6 |
| o3 (without search) | 49.4 | 43.2 | 14.0 | 14.0 | *48.9* | 33.9 | *20.3* | **88.9** | **85.4** | **95.4** | 72.5 |
| GPT-4o (with search) | *84.4* | *63.7* | 31.6 | 15.3 | 37.5 | *46.5* | 4.3 | *71.0* | *53.0* | *88.2* | 54.1 |
| o3 (with search) | **96.0** | **86.8** | 57.0 | 49.5 | 51.2 | 68.1 | **27.4** | **88.9** | **85.4** | **95.4** | 74.3 |
| *Open-Source Models* | | | | | | | | | | | |
| Qwen-2.5-7B | 4.0 | 16.5 | 2.1 | 1.4 | 6.2 | 6.0 | 1.2 | 10 | 33.0 | 61.2 | 24.7 |
| Qwen-2.5-7B + Search | 50.8 | 23.3 | 10.1 | 3.0 | 13.6 | 20.2 | 2.4 | 10 | 33.5 | 62.0 | 25.3 |
| Qwen3-4B | 3.8 | 14.7 | 2.6 | 2.1 | 9.0 | 6.4 | 4.2 | *65.0* | 55.1 | 71.0 | 48.8 |
| Qwen3-4B + Search | 67.7 | 27.2 | 17.5 | 6.2 | 18.7 | 27.5 | 6.2 | *65.0* | 55.9 | 72.0 | 49.8 |
| ZeroSearch-3B | 51.9 | 11.3 | 8.7 | 7.1 | 13.8 | 18.6 | 3.4 | 10.0 | 14.6 | 51.0 | 17.3 |
| ZeroSearch-7B | 75.3 | 30.0 | 18.2 | 6.2 | 20.6 | 30.1 | 4.2 | 10.0 | 29.3 | 57.5 | 22.8 |
| R1-Searcher-7B | 58.8 | 37.0 | 1.8 | 1.4 | 19.1 | 23.6 | 2.1 | 10.0 | 33.3 | 56.5 | 25.5 |
| search-o1 (Qwen3-4B) | 57.5 | 26.8 | 10.8 | 5.5 | 15.3 | 23.2 | 3.4 | 40.0 | 30.5 | 53.7 | 31.9 |
| WebSailor-3B | 87.1 | 44.4 | **52.2** | 9.0 | 27.4 | 44.0 | 7.4 | 40.0 | 45.5 | 51.3 | 36.0 |
| Jan-Nano-32K | 80.7 | 36.1 | 25.0 | 6.2 | 21.4 | 33.9 | 5.5 | 60.0 | 37.4 | 66.0 | 42.2 |
| Jan-Nano-128K | 83.2 | 43.4 | 33.7 | 6.2 | 23.9 | 38.1 | 6.1 | 53.3 | 51.0 | 65.4 | 44.0 |
| II-Search-4B | *88.2* | *58.7* | 40.8 | 17.1 | *31.8* | *47.3* | 7.4 | 60.0 | *51.5* | *72.1* | 47.8 |
| Fathom-Search-4B (Stage-1) | 88.1 | 57.2 | 39.0 | *19.8* | 31.3 | 47.1 | 6.7 | 60.0 | 55.6 | **75.4** | *49.4* |
| Fathom-Search-4B (Stage-2) | **90.0** | **64.8** | *50.0* | 22.5 | 33.2 | 52.1 | 9.5 | 70.0 | 60.1 | **75.4** | 53.8 |

Table 2: **Accuracy(%) of various Open/Closed-sourced DeepResearch-Agents and Search Augmented LLMs on DeepResearch-Bench.** Bold/italics denote best/second-best per category.

| Model | RACE | | | | | FACT | |
| --- | --- | --- | --- | --- | --- | --- | --- |
| | Overall | Comp. | Depth | Inst. | Read. | C. Acc. | E. Cit. |
| **Closed Source LLM with Search Tools** | | | | | | | |
| Claude-3.7-Sonnet w/Search | **40.67** | **38.99** | **37.66** | **45.77** | 41.46 | *93.68* | *32.48* |
| Perplexity-Sonar-Reasoning-Pro (high) | *40.22* | *37.38* | 36.11 | *45.66* | **44.74** | 39.36 | 8.35 |
| Gemini-2.5-Pro-Grounding | 35.12 | 34.06 | 29.79 | 41.67 | 37.16 | 81.81 | **32.88** |
| GPT-4o-Search-Preview (high) | 35.10 | 31.99 | 27.57 | 43.17 | 41.23 | 88.41 | 4.79 |
| GPT-4.1 w/Search (high) | 33.46 | 29.42 | 25.38 | 42.33 | 40.77 | 87.83 | 4.42 |
| **Closed Source Deep Research Agent** | | | | | | | |
| Grok Deeper Search | 40.24 | 37.97 | 35.37 | 46.30 | 44.05 | 83.59 | 8.15 |
| Perplexity-DeepResearch | 42.25 | 40.69 | 39.39 | 46.40 | 44.28 | **90.24** | 31.26 |
| Gemini-2.5-Pro DeepResearch | **48.88** | **48.53** | **48.50** | *49.18* | **49.44** | 81.44 | **111.21** |
| OpenAI-DeepResearch | *46.98* | *46.87* | *45.25* | **49.27** | *47.14* | 77.96 | *40.79* |
| **Open Source Deep Research Agent** | | | | | | | |
| Kimi-Researcher | *44.64* | **44.96** | *41.97* | 47.14 | *45.59* | – | – |
| Doubao-DeepResearch | 44.34 | 44.84 | 40.56 | 47.95 | 44.69 | *52.86* | **52.62** |
| LangChain Open-DeepResearch | 43.44 | 42.97 | 39.17 | *48.09* | 45.22 | – | – |
| Fathom-DeepResearch | **45.47** | *42.98* | **45.14** | **48.25** | **46.12** | **56.1** | *38.3* |

MedQA (Jin et al., 2021) train split. This combined pool is adversarially filtered against the Stage-1 checkpoint, yielding **5,471** instances. From MUSIQUE, we retain only questions requiring at least three reasoning hops to ensure sufficient compositional depth. For this stage, we adopt the Steerable Step-Level Reward (Eq. 6) to extend the tool-use horizon beyond 20 calls in a stable manner. We use the Qwen3-4B model (Yang et al., 2025) as the base, which supports a maximum context length of 40,960 tokens; we utilize the full window during training. We use GPT-4.1-mini(Temperature=0.) as the query LLM for training and evaluation unless stated otherwsie. A higher sampling temperature of 1.4 is applied to Qwen3 models, consistent with prior findings (An et al., 2025). All experiments are conducted on a single node with $8 \times$H100 GPUs.

## 3 FATHOM-SYNTHESIZER-4B

*Fathom-Synthesizer-4B* is a planning and synthesis model built on Qwen3-4B via supervised fine-tuning (SFT). It converts multi-hop DeepSearch traces from *Fathom-Search-4B* into decision-grade, citation-dense *DeepResearch Reports*. Following a *Plan-then-Write* protocol, the model first decomposes the question into sub-goals, defines the report structure, maps evidence to sections, and specifies strategies for insight generation; only then does it produce the public report with citations drawn strictly from URLs explored by *Fathom-Search-4B*. This explicit planning improves question alignment, strengthens citation accuracy through section-level constraints, and provides structured supervision during SFT, enhancing the distillation process.

### 3.1 DEEPRESEARCH-SFT

DEEPRESEARCH-SFT is a synthetic dataset distilled from GPT-5 (OpenAI, 2025a) to train *Fathom-Synthesizer-4B*, it provides supervision along three complementary axes: **(i) Question decomposition.** Each input question $q$ is decomposed into ordered sub-questions $\pi^{\text{decomp}} = (S_1, \ldots, S_n)$, which form the report scaffold and ensure coverage of all facets, **(ii) Section mapping.** Every piece of evidence recovered during search (URLs, quoted passages, tables, figures) is grounded to one or more sections via a mapping $\pi^{\text{map}}$, this aligns each explored URL to the most relevant $S_i$, enhancing citation accuracy and preventing omissions/duplication.**(iii) Planning for insights.** The model specifies an analysis strategy $\pi^{\text{insight}}$ how the gathered evidence should be synthesized into higher-level insights. Formally, given a question $q$ and trajectory $\tau = \{\mathcal{R}_1, \ldots, \mathcal{R}_T\}$, the teacher outputs `Plan` and `Report`. The plan $\pi = (\pi^{\text{decomp}}, \pi^{\text{map}}, \pi^{\text{insight}})$ appears in a private `<think>` block, followed by the public report $r$. The training target is $y = $ `<think>` $\pi$ `</think>` $r$.

**Report structure.** The public-facing report $r$ follows a fixed, inline-citation–driven format: an *Executive Summary* followed by a *Main Body* organized exactly by the sections $(S_1, \ldots, S_n)$ from $\pi^{\text{decomp}}$, where each section weaves the mapped evidence from $\pi^{\text{map}}$ using the analysis strategy in $\pi^{\text{insight}}$. Sections are citation-dense: every pivotal or non-obvious claim carries inline citations drawn *only* from URLs explored in $\tau$, with section-level citations restricted to items mapped to that section in $\pi^{\text{map}}$. The report concludes with a deduplicated *Sources used* list of the cited URLs.

**Thematic diversity & scale.** Training questions are generated via the *Seeded Question mode* (Sec 2.1), starting from 100 open-ended real-world questions spanning law, business, technology, science, and policy. Rewritten across sampled themes, this yields **2,500** questions for training. Refer to (Appendix A.2 for more details on the dataset)

### 3.2 TRAINING RECIPE

We fine-tune **Qwen3-4B** on DEEPRESEARCH-SFT, training for **5 epochs** on the 2,500-sample split using a single node of $8\times$H100 GPUs. We use `bf16`, FlashAttention-2, a 65,536-token context, gradient accumulation of 8, cosine LR with peak $5.0 \times 10^{-5}$, Adam ($\beta_1$=0.9, $\beta_2$=0.95), and sequence parallel size 4. **Context extension.** Our DeepSearch traces exhaust Qwen3-4B's native 40,960-token context window, so we *extend* the effective context during SFT using YaRN RoPE scaling: `rope_scaling:{type=yarn, factor=2.0}`. This increases the usable positional range to 65,536 tokens, allowing the synthesizer to ingest the full investigation trace and generate high-quality synthesis while preserving section alignment and citation locality.

## 4 BASELINES, BENCHMARKS & METRICS

**Baselines. Open-source** DeepSearch agents: with public checkpoints: Jan-Nano (Dao & Vu, 2025), II-Search-4B (Internet, 2025), Qwen3-4B (Yang et al., 2025), ZeroSearch (Sun et al., 2025a), Search-o1 (Li et al., 2025b), R1-Searcher (Song et al., 2025), WebSailor (Li et al., 2025a). **Closed-source:** comparators: o3 (OpenAI, 2025a), GPT-4o (OpenAI, 2024).

**Benchmarks (9). DeepSearch (5):** SimpleQA (Wei et al., 2024), FRAMES (Krishna et al., 2024), WebWalkerQA (Wu et al., 2025), Seal0 (Pham et al., 2025), MuSiQue (Trivedi et al., 2022). **General reasoning (4):** HLE (Phan et al., 2025), AIME-25 (AIME, 2025), GPQA-Diamond (Rein et al., 2024), MedQA (Jin et al., 2021). *Metric:* Pass@1 using GPT-4.1-mini LLM as Judge (Temperature=0).

Table 3: **Ablation** on using RAPO as the policy optimization algorithm for Stage-1 training compared to GRPO. Both trainings done on top of Qwen3-4B model.

| Algorithm | SimpleQA | FRAMES | WebWalker | Seal0 | Avg. Tokens |
|-----------|----------|--------|-----------|-------|-------------|
| GRPO | 87.8 | 55.2 | 33.8 | 14.4 | 9,000 |
| RAPO | 88.1 | 57.2 | 39.0 | 19.8 | 5,000 |

Table 4: **Ablation** on using our steerable step-level reward compared to the vanilla trajectory-level RLVR reward for Stage-2 training. Trained on top of Fathom-Search-4B (Stage-1) using RAPO.

| Reward | SimpleQA | FRAMES | WebWalker | Seal0 | Avg. Tokens |
|--------|----------|--------|-----------|-------|-------------|
| Vanilla Reward (Eq. 3) | 88.2 | 58.2 | 43.2 | 21.6 | 5,500 |
| Steerable Step-Level Reward (Eq. 6) | 90 | 64.8 | 50 | 22.5 | 14,500 |

**DeepResearch (1):** DeepResearch-Bench (Du et al., 2025). *Metrics:* RACE (reference-based adaptive criteria-driven evaluation of comprehensiveness, depth, instruction-following, readability) and FACT (factuality via citation accuracy and effective citation count) for open-ended citation driven report generation.

## 5 DISCUSSION

**Strong performance rivaling closed-source proprietary models** Fathom-DeepResearch establishes itself as a clear state-of-the-art by achieving large, non-incremental gains on the most challenging DeepSearch tasks like *FRAMES, WebWalker, & Seal0*, (Table. 1), while also showing strong generalization to broader reasoning benchmarks like (GPQA-Diamond and Humanity's Last Exam). Unlike many search-augmented systems that falter outside their training domain, it consistently outperforms both its base model and other open-source systems, and even surpasses larger closed-source models such as GPT-4o with notable margins. On open-ended benchmark: *DeepResearch-Bench*, it outperforms most proprietary closed-source systems (including Claude, Grok, and Perplexity Deep Research) (Table. 2) underscoring its competitiveness in end-to-end deep research tasks.

**On policy optimization: RAPO vs. GRPO.** Table. 3 contrasts RAPO and GRPO as the policy-optimization algorithm with the Stage-1 setup fixed. RAPO consistently outperforms GRPO across DeepSearch benchmarks, This shows that RAPO provides a more stable and effective optimization signal. As shown in Fig. 4, GRPO expands response length as training progresses, but this growth does not translate into higher accuracy because the model collapses into redundant tool-call spamming behavior. RAPO, in contrast, achieves stronger tool-calling efficiency and more accurate results.

**On the Steerable Step-Level reward.** As shown in Fig. 3, the steerable step-level reward provides a finer-grained training signal that steers the tool calling behavior of the model. By directly shaping the utility of each intermediate step, it encourages controlled growth in response length without inflating reasoning traces with redundant tool call spam, thereby yielding both efficiency and stability in multi-step reasoning, outperforming the vanilla RLVR reward function on all DeepSearch tasks as shown in Table. 4

**Limitations.** While RAPO is effective for stabilizing multi-turn RL training, it shows limited test-time scaling. As illustrated in Fig. 4, RAPO with vanilla rewards during Stage-2 training saturates before 6,000 tokens and yields only marginal accuracy

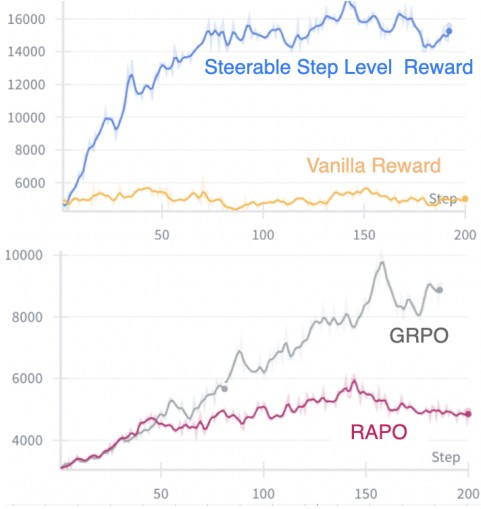

Figure 4: Response length evolution during **(i) top.** Stage-2 training (Steerable Step Level Reward vs. Vanilla Reward) & **(ii) bottom.** Stage-1 training (GRPO vs RAPO)

gains (Table 4) as question difficulty increases, when the steerable step-level reward is absent. This trade-off arises from its reliance on trajectory replacement in the replay buffer, which anchors learning to low-entropy traces which prevents training collapse but also hinders adaptation to extended reasoning horizons. More broadly, our current system depends on synchronous training pipelines that, although simple to implement, remain inefficient and brittle at scale. Transitioning to asynchronous frameworks presents a natural next step for improving efficiency and robustness.

## 6 CONCLUSION

We present Fathom-DeepResearch, an agentic system that addresses critical gaps in open-source deep research capabilities through two specialized 4B models: Fathom-Search-4B for multi-turn web search and reasoning, and Fathom-Synthesizer-4B for structured report synthesis. Our key contributions include DuetQA, a multi-agent self-play dataset that ensures search dependency; RAPO, a stabilized extension of GRPO that enables reliable tool use beyond 20 calls through curriculum pruning, advantage scaling, and replay buffers; a steerable step-level reward system that mitigates reward hacking while providing explicit control over exploration and verification behaviors; and DeepResearch-SFT, a synthetic corpus that enables comprehensive information synthesis through explicit plan-then-write supervision.

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

# A APPENDIX

## A.1 AGENTIC TOOL DESIGN

We provide our policy model access to two tools:

**search_urls (web search).** The tool takes as input a natural language query $q$ and returns a ranked list of triples $(u, \texttt{title}, \texttt{snippet})$ using a live search engine. The policy model uses this to identify promising sources and optionally select a URL $u$ for opening in the next step. The tool is invoked as follows: `<tool_call>{name: search_urls, args: {query: q}}</tool_call>`

**query_url (goal-conditioned page reading).** Given a goal $g$ and a URL $u$, the tool leverages a query LLM to return targeted evidence-backed response that address $g$. This tool enables precise grounding of facts and targeted querying of web-pages. Compared to the injestion of entire web-page into the policy model's trajectory, this tool minimizes noise and increases recall. The tool is invoked as follows: `<tool_call>{name: query_url, args: {goal: g, url: u}}</tool_call>`

## A.2 DATASETS

Datasets provided in the supplementary material:

**Stage-1 DeepSearch Training (DuetQA):** QA pairs (search-essential, multi-hop).

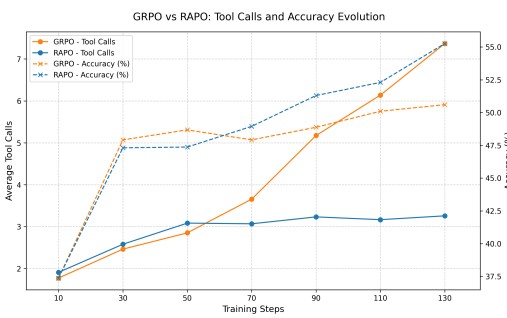

Figure 5: GRPO vs RAPO: Tool call and accuracy evolution

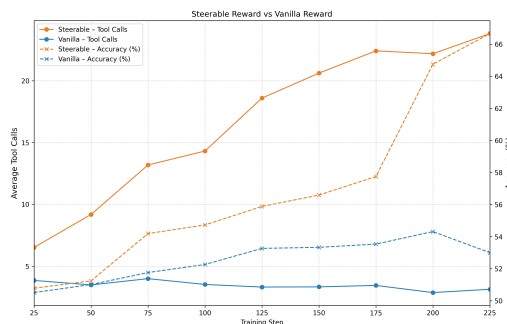

Figure 6: Steerable Reward vs Vanilla: Tool call and accuracy evolution

**Stage-2 DeepSearch + General Reasoning Training:** mixed QA pairs spanning DeepSearch and general reasoning tasks from S1 (maths)(Muennighoff et al., 2025), Musique (Trivedi et al., 2022)(multi-hop) and DuetQA(live-web-search), MedQA(Jin et al., 2021)(medical-reasoning) adversely filtered against Fathom-Search-Stage1 checkpoint.

**DeepResearch-SFT:** Contains synthetically generated open-ended questions, their DeepSearch traces from Fathom-Search-4B, the per-example planning used for report synthesis, and the final DeepResearch reports generated by GPT-5 from the corresponding search traces.

## A.3 TRAINING

From 5, GRPO shows rising tool use without accuracy gains (inefficient exploration), whereas RAPO achieves higher accuracy with fewer calls, indicating productive long-horizon reasoning.

From 6, With Steerable reward, accuracy increases monotonically with tool-call depth, indicating our Steerable reward leads to productive exploration, where increasing number of tool calls translate to higher accuracy, a signature of genuine long-horizon scaling absent in the vanilla reward.

From 7, Test-time scaling plot for Fathom-Search-4B's GRPO-trained checkpoint, RAPO-trained checkpoint, and WebSailor-3B (Li et al., 2025a). It clearly demonstrates that WebSailor-3B and GRPO saturate early ( 15 calls) as their performance does not increase with further tool calling , while our models continue improving as the call budget increases upto 30 tool calls, demonstrating superior scaling under long-horizon budget.

## A.4 FAILURE MODE ANALYSIS

To contextualize when long-horizon tool use is required and when models fail, we follow the task-uncertainty taxonomy introduced by WebSailor (Li et al., 2025a)

- **Level-1** tasks exhibit low uncertainty.
- **Level-2** tasks (e.g., multi-hop QA) begin with higher uncertainty but follow a deterministic, well-structured reasoning path in which uncertainty monotonically contracts as the model links successive entities; these questions can be solved with $\leq 25$ searches.
- **Level-3** DeepSearch tasks: the most challenging category combine high uncertainty at each step with no predefined reasoning path. Several intermediate hypotheses remain viable for many hops, and only cross-verification across distant, independent clues collapses the ambiguity to a correct answer. These questions require up to 100 searches.

Level-1 tasks are oftens solved by model's pretrained knowledge or 1–2 searches.

Level-2 tasks (e.g., multi-hop QA) begin with higher uncertainty but follow a deterministic, well-structured reasoning path in which uncertainty monotonically contracts as the model links successive entities; these questions can be solved with $\leq 25$ searches.

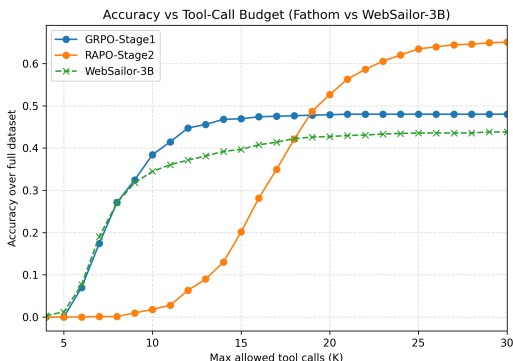

Figure 7: Test-Time Scaling comparison of RAPO vs GRPO vs WebSailor-3B

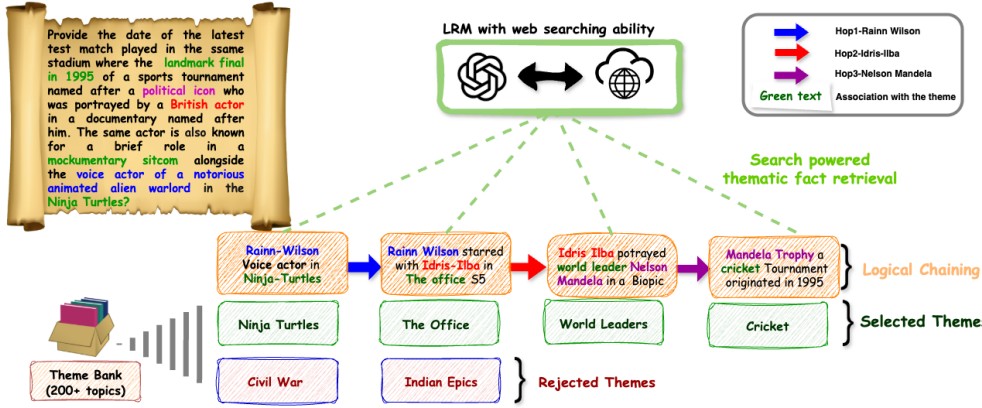

Figure 8: Sample question from DuetQA, generated using the *Mixture-of-Themes mode*

Our present issues stem not from the method but from the context budget. We train Qwen3-4B (Yang et al., 2025) at a 40,960-token context window (no RoPE/YARN extension) due to compute constraints, which caps feasible trajectories at $\leq 30$ tool calls. Level-3 DeepSearch tasks (Wei et al., 2025) however, routinely require 50–100+ calls and 128K–256K context, making them impossible to model faithfully under our current budget.

Equally important, Level-3 problems require fuzzy, high-entropy training data, where multiple intermediate hypotheses remain viable for many hops and uncertainty is intentionally preserved rather than collapsing early. While WebSailor (Li et al., 2025a) reports using such ambiguity-preserving supervision, their dataset and generation pipeline are not open-sourced, so it is unclear how these high-uncertainty questions are produced for training. In contrast, we open-source our entire data-generation pipeline and will extend it in the next iteration to explicitly produce Level-3–style, multi-hypothesis, long-horizon training data once we can train with a 128K-context window.

# B ACKNOWLEDGEMENT

## B.1 USE OF LLMS

We acknowledge the use of LLMs for basic writing, grammar and formatting enhancements in limited capacity within the paper.

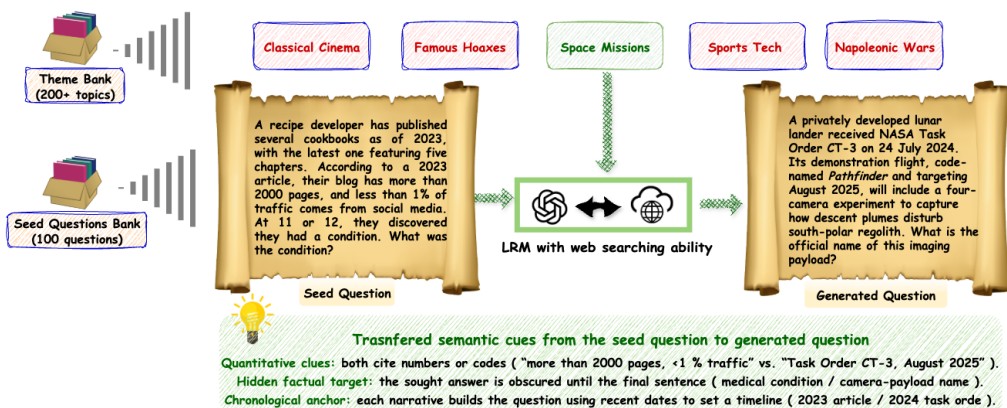

Figure 9: Sample question from DuetQA, generated using the *Seeded-Question-Generation mode*

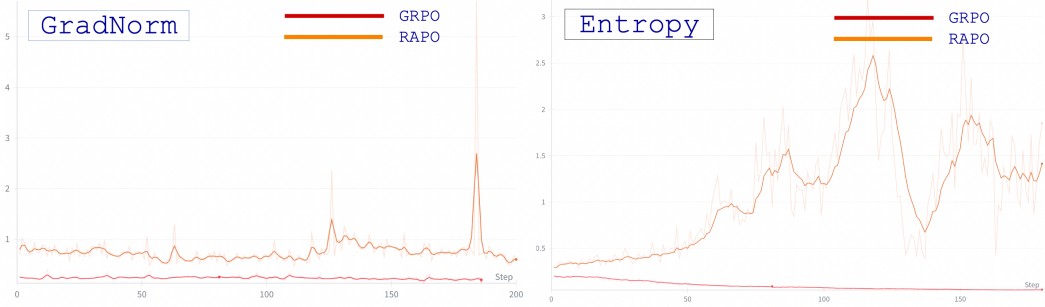

Figure 10: Comparison of policy entropy and gradient norm during RLVR training. GRPO exhibits rapid entropy collapse and diminished gradient norms due to sparse rewards, whereas RAPO sustains exploration and stronger updates via targeted updates

