# OpenReview forum: "Fathom-DeepResearch: Unlocking Long Horizon Information Retrieval and Synthesis for SLMs"
_ICLR.cc/2026/Conference — ICLR 2026 Poster_

### Official Review · Reviewer_CAQE · 2025-10-26

**Soundness:** 3
**Presentation:** 3
**Contribution:** 3
**Rating:** 6
**Confidence:** 3

**Summary:**

This paper presents Fathom-DeepResearch, an open-source framework for long-horizon information retrieval and data synthesis for deep research agents.

**Strengths:**

1.	The motivation is clear and well-grounded, directly addressing key limitations in current DeepResearch models, such as unstable multi-turn reinforcement learning and inefficient tool usage.
2.	The proposed methodologies are sound and well-justified, with each component, RAPO, DUETQA, and the Steerable Step-Level Reward, clearly contributing to overall model stability and performance.
3.	The experiments are comprehensive, spanning both DeepResearch-specific and general reasoning benchmarks, and the results convincingly demonstrate the framework’s effectiveness and competitiveness against baselines.

**Weaknesses:**

1. The paper points out that GRPO suffers from reward-hacking behaviors, where the model tends to overuse tools without improving reasoning quality. However, it does not clearly demonstrate that the proposed RAPO algorithm effectively mitigates this issue from the perspective of actual tool-call frequency. In Figure 4, only the response length is presented as a proxy for the number of tool calls, which is an indirect measure. A more concrete quantitative analysis or visualization of tool-call usage would strengthen the claim that RAPO truly alleviates reward-hacking behavior.
2. More challenging  benchmarks, such as BrowseComp, are expected to be included to better demonstrate the model’s deep research capabilities and generalization to new, open-ended evaluation settings.

**Questions:**

1. What's the meaning of "SLMs"? The title mentions SLMs, but the paper does not define this abbreviation.

---

> ### Author Response · Authors · 2025-11-20
> **Respone 1/n**
>
> # Reviewer-CAQE
>
> **[Weakness]** The paper points out that GRPO suffers from reward-hacking behaviors, where the model tends to overuse tools without improving reasoning quality. However, it does not clearly demonstrate that the proposed RAPO algorithm effectively mitigates this issue from the perspective of actual tool-call frequency. In Figure 4, only the response length is presented as a proxy for the number of tool calls, which is an indirect measure. A more concrete quantitative analysis or visualization of tool-call usage would strengthen the claim that RAPO truly alleviates reward-hacking behavior.
>
> **[Ans]** We thank the reviewer for raising this point. Here we provide 2 additional plots that compare the evolution of accuracy and tool-call depth over the course of stage-1 & stage-2 of training. This serves as a direct measure of the effectiveness of our test-time scaling methods and alleviations from “reward-hacking” and "tool-spamming" behavior.
>
> **Fig. 1 (GRPO vs RAPO):** Tool call and accuracy evolution on the validation set over stage-1 training, GRPO shows rising tool use without accuracy gains (inefficient exploration), whereas RAPO achieves higher accuracy with fewer calls, indicating productive long-horizon reasoning. [ https://anonymous.4open.science/r/Rebuttal-stuff-2B00/stage_1_final.png ]
>
> **Fig. 2 (Steerable Reward vs Vanilla):** Tool call and accuracy evolution on the validation set over stage-2 training. With Steerable reward, accuracy increases monotonically with tool-call depth, indicating that our Steerable reward leads to productive exploration, where increasing number of tool calls translate to higher accuracy, a clear demonstration of genuine long-horizon scaling absent in the vanilla reward.[ https://anonymous.4open.science/r/Rebuttal-stuff-2B00/stage_2_final.png ]
>
> Note: The validation set is a set of 200 question randomly sampled from the frames dataset.
>
> ---
> **[Question]** More challenging benchmarks, such as BrowseComp, are expected to be included to better demonstrate the model’s deep research capabilities and generalization to new, open-ended evaluation settings.
>
>
> **[Ans]** We appreciate the reviewer’s suggestion to evaluate **Fathom** on the most challenging DeepSearch benchmark **BrowseComp**. We originally did not include **BrowseComp** because it cannot be meaningfully evaluated under our current training and context-budget constraints. **BrowseComp** queries typically require **50–100+ tool calls** and **128K–256K** context windows, whereas our model is trained on **Qwen3-4B** with a **40,960-token** window (no RoPE/YARN extension), limiting feasible trajectories to **<30 calls**. Under these conditions, the benchmark becomes a test of context limits rather than the underlying method.
>
> As requested, we now provide a comparison of **Fathom-Search-4B** on **BrowseComp** against models of similar size and scope [ https://anonymous.4open.science/r/Rebuttal-stuff-2B00/browsecomp_passk_rebuttal.png ]. **Fathom-Search-4B** **outperforms strong open-source baselines** such as **II-Search-4B**, **WebSailor-3B**, and **Jan-Nano-128K** on **Pass@1**, and its **Pass@5** performance is already **competitive with GPT-oss-20B (high)** despite the significant model-size and context-budget gap. This indicates that our method scales favorably even in a setting for which it was not explicitly optimized.
>
> Pass@1 comparison
>
> | Model              | Pass@1 |
> |--------------------|--------|
> | Fathom-Search-4B   | 3.90   |
> | II-Search-4B       | 3.40   |
> | WebSailor-3B       | 3.30   |
> | GPT-oss-20B (low)  | 2.80   |
> | Jan-Nano-128K      | 1.70   |
>
> #### Pass@5 comparison
>
> | Model                          | Score |
> |--------------------------------|-------|
> | Fathom-Search-4B (Pass@5)      | 7.82  |
> | GPT-oss-20B (high) (Pass@1)    | 8.20  |
>
> ---
>
> **[Question]** What's the meaning of "SLMs"? The title mentions SLMs, but the paper does not define this abbreviation.
>
> **[Ans]** We apologize for not making it clear in the manuscript. **SLMs** stand for **Small Language Models** and is used to refer to models with parameter range in the low billions or millions. We will update this in the manuscript.

---

> > ### Author Response · Authors · 2025-11-26
> > **Gentle reminder**
> >
> > Dear Reviewer:
> > We hope you are doing well. We wanted to express our sincere thanks for the time and effort you have already invested in reviewing our submission. We appreciate the thoughtful feedback and have carefully addressed each point in our rebuttal.
> > We are writing to kindly inquire whether there are any remaining concerns or clarifications that would assist you in completing your evaluation. If there are aspects where additional explanation, experiments, or evidence would be helpful, we would be happy to provide them promptly. Our intention is simply to ensure that our responses fully resolve the issues raised and support you in forming an informed and confident final assessment. Please do let us know if there is anything further we can clarify that would help strengthen our case.
> > Thank you once again for your time and consideration.

---

### Official Review · Reviewer_ms6f · 2025-10-31

**Soundness:** 3
**Presentation:** 2
**Contribution:** 3
**Rating:** 6
**Confidence:** 3

**Summary:**

This paper presents a two-part agentic reasoning system built from Qwen3-4B models: Fathom-Search-4B, which performs multi-turn, evidence-based web investigation, and Fathom-Synthesizer-4B, which transforms search traces into citation-dense reports. The system introduces three contributions — the synthetic DUETQA dataset for training, the RAPO algorithm for stable multi-turn reinforcement learning, and a steerable step-level reward.  This work is evaluated across multiple benchmarks, including DeepSearch and general reasoning benchmarks, on both closed-source and open-source models/agents.

**Strengths:**

1. This work provides synthetic training data to facilitate training the deep research agents.
2. This work proposes reward-aware policy optimization (RAPO), and the experiments show the proposed methods can achieve better performance than vanilla GRPO.
3. This work provides extensive experiments to verify the effectiveness of the method on multiple benchmarks and compare different closed-source and open-source baselines.

**Weaknesses:**

1. This work lacks a detailed analysis of whether the final trained model can truly work for long-horizon tool calls. When it succeeds and when it fails?

**Questions:**

1. How long does it take to train the model for different stages?
2. This work is claimed to build on top of RECALL (Chen et al., 20). Why not treat it as one of the baselines?

---

> ### Author Response · Authors · 2025-11-20
> **Response 1/n**
>
> # Reviewer-ms6f
>
> **[Weakness]** This work lacks a detailed analysis of whether the final trained model can truly work for long-horizon tool calls. When it succeeds and when it fails?
>
> **[Ans]**    We thank the reviewer for raising this point. Our model does perform substantive long-horizon tool-based reasoning, and we provide both empirical evidence for the same and a principled task taxonomy to clarify when it succeeds and when it fails.
>
> **Demonstrated long-horizon capability**
>
>   Despite using a **40,960-token** context window (**Qwen3-4B**, without RoPE/YARN due to resource limits), our system can reliably executes **~30 sequential tool calls per trajectory**. This is already significantly stronger than the best open-source baselines for ≤4B model size, like **II-Search**, **Jan-Nano**, and others, despite their larger (**128K**) context lengths. Please refer to Fig. 1 in the manuscript for exact comparison.
>
> **Evidence of efficient long-horizon reasoning (Figs. 1–3)**
>
>    - **Fig. 1 (GRPO vs RAPO):** Tool call and accuracy evolution on the validation set over stage-1 training. **GRPO** shows rising tool use without accuracy gains (inefficient exploration), whereas **RAPO** achieves higher accuracy with fewer calls, indicating productive long-horizon reasoning.[ https://anonymous.4open.science/r/Rebuttal-stuff-2B00/stage_1_final.png ]
>
>
>    - **Fig. 2 (Steerable Reward vs Vanilla):** Tool call and accuracy evolution on the validation set over stage-2 training. With **Steerable reward**, accuracy increases monotonically with tool-call depth, indicating our Steerable reward leads to productive exploration, where increasing number of tool calls translate to higher accuracy, a signature of genuine long-horizon scaling absent in the vanilla reward.[ https://anonymous.4open.science/r/Rebuttal-stuff-2B00/stage_2_final.png ]
>
>
>    - **Fig. 3 (Test-Time Scaling vs WebSailor-3B):** Test-time scaling plot for **Fathom-Search-4B’s** GRPO-trained checkpoint, RAPO-trained checkpoint, and **WebSailor-3B** (Li et. al.). It clearly demonstrates that **WebSailor-3B** and **GRPO** saturate early (~15 calls) as their performance does not increase with further tool calling , while our models continue improving as the call budget increases upto **30 tool calls**, demonstrating superior scaling under long-horizon budget.[ https://anonymous.4open.science/r/Rebuttal-stuff-2B00/toolcall_scaling_fathom_vs_websailor.png ]
>
> 3. **Failure Modes**
>
>    To contextualize when long-horizon tool use is required and when models fail, we follow the task-uncertainty taxonomy introduced by **WebSailor** (Li et. al.).
>
>    - **Level-1 tasks** exhibit low uncertainty and can be solved using internal knowledge or 1–2 searches.
>
>    - **Level-2 tasks** (e.g., multi-hop QA) begin with higher uncertainty but follow a deterministic, well-structured reasoning path in which uncertainty monotonically contracts as the model links successive entities; these questions can be solved with **≤25 searches**.
>
>    - **Level-3 DeepSearch tasks:** the most challenging category combine high uncertainty at each step with no predefined reasoning path. Several intermediate hypotheses remain viable for many hops, and only cross-verification across distant, independent clues collapses the ambiguity to a correct answer. These questions require **up to 100 searches**.
>
>    Our present issues stem not from the method but from the context budget. We train **Qwen3-4B** at a **40,960-token** context window (no RoPE/YARN extension) due to compute constraints, which caps feasible trajectories at **<30 tool calls**. **Level-3 DeepSearch tasks** (e.g., **BrowseComp**), however, routinely require **50–100+ calls** and **128K–256K** context, making them impossible to model faithfully under our current budget.
>
>    Equally important, Level-3 problems require fuzzy, high-entropy training data, where multiple intermediate hypotheses remain viable for many hops and uncertainty is intentionally preserved rather than collapsing early. While **WebSailor** reports using such ambiguity-preserving supervision, their dataset and generation pipeline are not open-sourced, so it is unclear how these high-uncertainty questions are produced for training. In contrast, we open-source our entire data-generation pipeline and will extend it in the next iteration to explicitly produce Level-3–style, multi-hypothesis, long-horizon training data once we can train with a **128K-context window**.

---

> > ### Author Response · Authors · 2025-11-20
> > **Response 2/n**
> >
> > # Reviewer-ms6f
> >
> > **[Question]** How long does it take to train the model for different stages?
> >
> > **[Ans]**  Single node with 8×H100 GPUs (3 days for stage-1, 2 days for stage-2). We plan on integrating modern asynchronous frameworks to speedup training and scale to multiple nodes.
> >
> > ---
> >
> > **[Question]** his work is claimed to build on top of RECALL (Chen et al., 20). Why not treat it as one of the baselines?
> >
> > **[Ans]**   RECALL (Chen et al.) is a RAG-style framework whose retriever operates over a static, pre-2024 Wikipedia index. Our experiments are explicitly designed to build models capable of live-web search over heterogeneous web sources beyond Wikipedia, and to enforce dependence on post-2024 information. Because RECALL cannot retrieve beyond its fixed Wikipedia snapshot and therefore cannot satisfy the temporal or source-diversity constraints central to our evaluation, it is not an appropriate baseline for our setting.

---

> > > ### Author Response · Authors · 2025-11-26
> > > **Gentle Reminder**
> > >
> > > Dear Reviewer: We hope you are doing well. We wanted to express our sincere thanks for the time and effort you have already invested in reviewing our submission. We appreciate the thoughtful feedback and have carefully addressed each point in our rebuttal. We are writing to kindly inquire whether there are any remaining concerns or clarifications that would assist you in completing your evaluation. If there are aspects where additional explanation, experiments, or evidence would be helpful, we would be happy to provide them promptly. Our intention is simply to ensure that our responses fully resolve the issues raised and support you in forming an informed and confident final assessment. Please do let us know if there is anything further we can clarify that would help strengthen our case. Thank you once again for your time and consideration

---

### Official Review · Reviewer_nMq5 · 2025-11-01

**Soundness:** 3
**Presentation:** 2
**Contribution:** 3
**Rating:** 6
**Confidence:** 3

**Summary:**

This paper introduces Fathom-DeepResearch, an open-source deep research agent system. The system consists of two coordinated 4B-parameter components: Fathom-Search-4B, a web-search model trained with an improved stabilized reinforcement-learning framework, and Fathom-Synthesizer-4B, a structured report generator trained under a *plan-then-write* protocol. The paper further introduces a dataset (DUETQA ) of 5k self-play-generated, live-search-dependent QA examples, and a synthetic DeepResearch-SFT corpus for training long-form synthesis. Evaluations across benchmarks show competitive performance of the proposed agent.

**Strengths:**

The work is well-motivated and technically complete, with reasonable contributions in this crowded space (1) a simple multi-agent framework and models (while most works explore single end-to-end model) (2) proposing a new RL training framework (3) datasets

**Weaknesses:**

- As this is a quickly evolving and crowded space, it's better to compare with more recent baselines, and more importantly, to compare with more recent related research on data synthesis and recent upgrades to GRPO (such as GSPO which is now widely used in this space).
- Relatedly, the components in this work, including the data synthesis and minor improvements on the training part, by far, have been largely explored, which undermines the contribution of this work.
- The RAPO relative to GRPO is still minor update per se, and the improvements in Table 3 is not very substantial.
- To thoroughly evaluate the "deep research" capabilities of the agent, the paper would be benefited from incorporating commonly used browsecomp benchmark. Some RAG benchmarks such as SimpleQA is already relatively too old and too short to be informative in this space.

**Questions:**

1. A typo in the Table 2: systems such Kimi-Researcher should be closed source agents?
2. Will all the models and datasets be opensourced?
3. Broader comparison:
   1. The authors might consider summarizing how Fathom differs from other recent research agents (in dataset construction, training paradigm, and report synthesis) to contextualize its contributions within the current surge of deep-research systems.
   2. the same for RL algorithms, such as GSPO which is now widely used by search agents.

---

> ### Author Response · Authors · 2025-11-20
> **Response 1/n**
>
> # Reviewer-nMq5
>
> **[Question]** A typo in the Table 2: systems such **Kimi-Researcher** should be closed source agents?
>
> **[Ans]** We thank the reviewer for pointing it out, we apologize for the oversight we will fix it in the manuscript placing Kimi-Researcher in the closed-source section.
>
> ---
>
> **[Question]**  Will all the models and datasets be open sourced?
>
> **[Ans]**  Yes, all the models and datasets and dataset generation code will be open sourced.
>
> ---
>
> **[Weakness]** Relatedly, the components in this work, including the data synthesis and minor improvements on the training part, by far, have been largely explored, which undermines the contribution of this work. The **RAPO** relative to **GRPO** is still minor update per se, and the improvements in Table 3 is not very substantial.
>
> [&]
>
> **[Question]** The authors might consider summarizing how **Fathom** differs from other recent research agents (in dataset construction, training paradigm, and report synthesis) to contextualize its contributions within the current surge of DeepResearch systems.
>
>
> **[Ans]** We appreciate the reviewers' concern about the distinguishing of our work in the recent surge of DeepResearch systems. To this regard we put forth the following arguments.
>
> ### 1. Dataset construction
>
> Unlike recent DeepSearch dataset pipelines such as **WebSailor** (Li et al.) and **ASearcher** (Gao et al.), **Fathom** adopts a different data-generation design. As shown in Sec. 2.1 and Fig. 4, **DUETQA** enforces (i) live-web search dependency by ensuring that at least one reasoning hop requires post-2024 evidence (ii) deliberate heterogeneous source diversification across non-Wikipedia domains, this design minimizes pre-training overlap and prevents models from short-circuiting multi-hop reasoning.
> Further, **DUETQA** is operationally simple: each question is produced using only three model calls (one for generation and two for verification). This contrasts sharply with **ASearcher** and **WebSailor**, which rely on multi-step page traversal and multi-turn retrieval chains to synthesize each question. Notably **WebSailor** has not released its full dataset and data generation code while we intend to fully open source our work.
> Finally, **DUETQA** provides explicit thematic via mixture-of-themes mode through a manually curated and editable taxonomy of 200+ themes and question framing style control via seeded question mode, enabling fine-grained steering of question domains and styles. Such steerability is absent in other approaches.
>
> | Work       | Live-web-search | Dataset Steerability    | Dataset open source | Code open source |
> | ---------- | ---------------- | ----------------------- | ------------------- | ---------------- |
> | WebSailor  | No               | No                      | No                  | No               |
> | ASearcher  | No               | No                      | Yes                 | Yes              |
> | SFR-DR     | No               | No                      | No                  | No               |
> | DUETQA     | Yes              | Yes                     | Yes                 | Yes              |
>
> ### 2. Report synthesis
>
> As detailed in Sec. 3.1, **Fathom** introduces, to our knowledge, the first 4B-scale DeepResearch system capable of producing **structured**, **citation-grounded** research reports to **open-ended questions**. Very few open-source models like **SFR-DR** (Nguyen et. eal), **Tongyi-DeepResearch** (Li et. al.) possess open-ended research synthesis capability, and none do so at this small (4B) scale and scope.
> Our **DEEPRESEARCH-SFT** dataset is likewise the first of its kind, providing supervision along three orthogonal axes: (i) question decomposition into ordered, semantically coherent sections; (ii) evidence-to-section alignment to ensure citation accuracy; and (iii) insight-planning to guide the synthesis of higher-level claims.
>
> ### 3. Training paradigm
>
> As mentioned in Sec. 1.2, **Fathom** introduces a two-stage **RL-Zero** framework that eliminates the cold-start phase used in all prior DeepResearch agents. Cold-start imprints strong bias in the cognitive behaviors instilled in the student model, causing downstream model behavior to depend heavily on the quality of that cold-start data. In contrast, our RL-Zero pipeline learns long-horizon tool-use in a steerable manner, without inheriting any behavioral bias from a handcrafted or imperfect cold-start set. The stepwise steerable reward provides explicit, fine-grained control over the model’s evolving reasoning trajectory, allowing control over the acquired cognitive behaviors such as exploration verification from first principles rather than from pre-imposed patterns.

---

> > ### Comment · Reviewer_nMq5 · 2025-11-20
> >
> > a quick comment: Kimi K2 is opensourced. Kimi-Researcher is close-source. You might as well double check what was tested in the exp.

---

> ### Author Response · Authors · 2025-11-20
> **Response 2/n**
>
> # Reviewer-nMq5
>
> **[Weakness]** As this is a quickly evolving and crowded space, it's better to compare with more recent baselines, and more importantly, to compare with more recent related research on data synthesis and recent upgrades to **GRPO** (such as **GSPO** which is now widely used in this space).
>
> [&]
>
> **[Question]** The same for RL algorithms, such as **GSPO** which is now widely used by search agents.
>
> **[Ans]** We appreciate the reviewers' concern about RAPO's comparison with GSPO, and present the following explanation.
>
> **GSPO** is concurrent work: the paper (`arXiv:2507.18071`) was released on 25th July, after our literature review cutoff in July, and thus it was not included in our experiments. We will be happy to mark this clearly in the paper and add **GSPO** to the related-work section.
>
> **GSPO** enhances **GRPO** by replacing token-level importance weighting with sequence-level ratios and clipping, yielding improved stability for multi-step tool-calling, particularly in MoE architectures. **RAPO**, by contrast, targets a distinct failure mode: group-variance collapse (σ→0) that emerges in multi-turn, tool-augmented reasoning. While **RAPO** itself does not induce long-horizon tool-use behavior, its combination with our step-wise steerable reward provides a principled mechanism for expanding and regulating multi-step tool-calling trajectories through cognitively controlled reward shaping. Crucially, **RAPO** is designed to be architecturally lightweight and complementary to all **GRPO** variants, including **GSPO**; however, due to compute constraints, we will be unable to train a full **GSPO+RAPO** system during the rebuttal period.
>
> ---
>
> **[Weakness]** To thoroughly evaluate the "deep research" capabilities of the agent, the paper would be benefited from incorporating commonly used **BrowseComp** benchmark. Some RAG benchmarks such as **SimpleQA** is already relatively too old and too short to be informative in this space.
>
> **[Ans]** We appreciate the reviewer’s suggestion to evaluate **Fathom** on the most challenging DeepSearch benchmark **BrowseComp**. We originally did not include **BrowseComp** because it cannot be meaningfully evaluated under our current training and context-budget constraints. **BrowseComp** queries typically require **50–100+ tool calls** and **128K–256K** context windows, whereas our model is trained on **Qwen3-4B** with a **40,960-token** window (no RoPE/YARN extension), limiting feasible trajectories to **<30 calls**. Under these conditions, the benchmark becomes a test of context limits rather than the underlying method.
>
> As requested, we now provide a comparison of **Fathom-Search-4B** on **BrowseComp** against models of similar size and scope [ https://anonymous.4open.science/r/Rebuttal-stuff-2B00/browsecomp_passk_rebuttal.png ]. **Fathom-Search-4B** **outperforms strong open-source baselines** such as **II-Search-4B**, **WebSailor-3B**, and **Jan-Nano-128K** on **Pass@1**, and its **Pass@5** performance is already **competitive with GPT-oss-20B (high)** despite the significant model-size and context-budget gap. This indicates that our method scales favorably even in a setting for which it was not explicitly optimized.
>
> Pass@1 comparison
>
> | Model              | Pass@1 |
> |--------------------|--------|
> | Fathom-Search-4B   | 3.90   |
> | II-Search-4B       | 3.40   |
> | WebSailor-3B       | 3.30   |
> | GPT-oss-20B (low)  | 2.80   |
> | Jan-Nano-128K      | 1.70   |
>
> #### Pass@5 comparison
>
> | Model                          | Score |
> |--------------------------------|-------|
> | Fathom-Search-4B (Pass@5)      | 7.82  |
> | GPT-oss-20B (high) (Pass@1)    | 8.20  |

---

> > ### Comment · Reviewer_nMq5 · 2025-11-20
> >
> > Good rebuttal. I will maintain my positive score here.

---

> ### Author Response · Authors · 2025-11-20
>
> We would like to thank the reviewer for their supportive feedback and positive response.
> Final clarification: We have used Kimi-Researcher API, we acknowledge the reviewers input and will place it in the closed-source category in the manuscript. The mention of Kimi-K2 in the previous rebuttal was a typing mistake, we apologize.

---

### Meta-Review · Area_Chair_zPYC · 2026-01-07

**Summary:**

This paper proposes two models trained from Qwen3-4B: Fathom-Search-4B and Fathom-Synthesizer-4B. The first model performs multi-turn web-search and the second generates a structured report using a “plan-then-write” protocol. These models are trained using their proposed RAPO algorithm, which provides more multi-turn stability than GRPO. The authors also open-source two synthetic datasets used for training: DuetQA and DeepResearch-SFT, for multi-turn web search and deep research report generation. They find that their models provide competitive performance on both tasks across several benchmarks within their parameter class.

These were the most important concerns from the reviewers but they addressed well by the authors. I expect all of these responses to be added to the paper prior to publication.

- Novelty is limited & RAPO is a minor update and Table 3 improvements are minor
	- 1) DuetQA is fully open source and allows for live-web search and more heterogeneous sources across non-Wikipedia domains when previous methods do not.
	- 2) The Fathom model is the smallest model to engage in DeepResearch and the DeepResearch-SFT dataset is the first of its kind.
	- 3)  Fathom eliminates the “cold-start phase” used in previous DeepResearch agents.

- Can Fathom actually handle long-horizon tool calling?
	- Added three new figures to show that Fathom can perform long-horizon tool calling effectively. I expect to see these figures added to the paper.
	- Failure modes are discussed but the authors do not mention that this will be added to the paper. I would like to see this discussion added to the main paper with some more detail on the performance on Level 1,2 and 3 tasks.

- SimpleQA is too old and short to be informative
	- Fathom shows similar performance trends in BrowseComp compared to other similarly sized models even though it was not optimized on it and its context length is too small to tackle it convincingly.

**Reviewer Concerns:**

R1: Reviewer accepted the rebuttal and kept the high score of 6
- Novelty is limited & RAPO is a minor update and Table 3 improvements are minor
	- 1) DuetQA is fully open source and allows for live-web search and more heterogeneous sources across non-Wikipedia domains when previous methods do not.
	- 2) The Fathom model is the smallest model to engage in DeepResearch and the DeepResearch-SFT dataset is the first of its kind.
	- 3)  Fathom eliminates the “cold-start phase” used in previous DeepResearch agents.
- Comparison with recent baselines (GSPO)
	- Concurrent work and RAPO is complementary to GSPO
- SimpleQA is too old and short to be informative
	- Fathom shows similar performance trends in BrowseComp compared to other similarly sized models even though it was not optimized on it and its context length is too small to tackle it convincingly.

R2:
- Detailed analysis on whether Fathom can handle long-horizon tool calling
	- Added three new figures to show that Fathom can perform long-horizon tool calling effectively. I expect to see these figures added to the paper.
	- Failure modes are discussed but the authors do not mention that this will be added to the paper. I would like to see this discussion added to the main paper with some more detail on the performance on Level 1,2 and 3 tasks.
- Why is RECALL not a baseline?
	- RECALL is based only on Wikipedia so it cannot be used directly.
R3:
- A more thorough analysis is needed to show whether RAPO overuses tools less than GRPO
	- Same response to R2’s concern about long-horizon tool calling
- More challenging benchmarks are necessary
	- Same response to R1 about SimpleQA being old and simple.

**Reviewer Scores:**

- nMq5 6 -> 6
- ms6f  6 -> 6
- CAQE 6 -> 6

The reviewers mainly wanted more evidence for the author’s claims and the rebuttal provided this in the form of more analysis and empirical results. I recommend that this paper be accepted.

---

### Decision · Program_Chairs · 2026-01-26

Accept (Poster)